# Psychometric Properties of Fears of Cancer Recurrence Scales in Turkish Hematologic Cancer Patients

**DOI:** 10.3390/medicina61091628

**Published:** 2025-09-09

**Authors:** Serkan Guven, Nursel Topkaya, Ertuğrul Şahin, Samet Yaman, Ufuk Demirci, Fatos Dilan Koseoglu

**Affiliations:** 1Department of Hematology, Çanakkale Mehmet Akif Ersoy State Hospital, 17100 Çanakkale, Türkiye; 2Department of Guidance and Psychological Counseling, Faculty of Education, Çanakkale Onsekiz Mart University, 17000 Çanakkale, Türkiye; nursel.topkaya@comu.edu.tr; 3Department of Guidance and Psychological Counseling, Faculty of Education, Amasya University, 05100 Amasya Merkez, Türkiye; ertugrulsahin@amasya.edu.tr; 4Department of Hematology, Hitit University Çorum Erol Olçok Training and Research Hospital, 19040 Corum, Türkiye; drsametyaman@hotmail.com; 5Department of Hematology, Medical Faculty, Manisa Celal Bayar University, 45140 Manisa, Türkiye; ufukdemirci3232@gmail.com; 6Department of Internal Medicine, Faculty of Medicine, Bakırçay University, 35665 Izmir, Türkiye; fatosdilankoseoglu@gmail.com

**Keywords:** hematological cancer patients, FCR-7, FCR-4, FCR-6, validity, reliability, Türkiye

## Abstract

*Background and Objectives*: Fears of cancer recurrence (FCR) represents one of the most common psychological problems in cancer patients. Therefore, valid and reliable measurement tools are needed to assess FCR in this population. The purpose of this study was to examine the psychometric properties of FCR scales (Fears of Cancer Recurrence-7 Item Version [FCR-7]; FCR-4 Item Version [FCR-4]; FCR-6 Item Version [FCR-6]) in Turkish hematological cancer patients. *Materials and Methods*: The study sample consisted of 239 hematological cancer patients undergoing treatment at four different state hospitals in Türkiye. *Results*: Confirmatory factor analysis results showed that all three scales had a single-factor structure (FCR-7: CFI = 0.981, TLI = 0.969, RMSEA = 0.071, SRMR = 0.028; FCR-4: CFI = 1.000, TLI = 1.001, RMSEA = 0.000, SRMR = 0.004; FCR-6: CFI = 0.981, TLI = 0.965, RMSEA = 0.087, SRMR = 0.028). The model allowing for correlated error terms between the first and second items provided the best fit. Research findings also indicated that the scales possessed strict measurement invariance across gender. Convergent and discriminant validity analyses also demonstrated expected associations between FCR scale scores and psychological well-being (*r* = −0.25 to −0.34) and psychological distress (*r* = 0.33 to 0.46) variables. The known-groups validity analysis indicated that the scales were effective at differentiating between groups and that they replicated the well-established finding from previous research that women report significantly higher levels of FCR than men (*d* = 0.42–0.47). Reliability analyses indicated that all three versions of the FCR scales had high internal consistency (Cronbach’s α = 0.91–0.93) and strong test–retest reliability (*r* = 0.85–0.87). *Conclusions*: Overall, the findings indicated that FCR-7, FCR-4, and FCR-6 are psychometrically sound, valid, and reliable instruments for assessing FCR in Turkish hematological cancer patients. These instruments can be used in clinical practice and research studies, as well as for evaluating intervention effectiveness in this population.

## 1. Introduction

Cancer is one of the most significant public health challenges countries face and a leading cause of morbidity and mortality that affects an increasing number of people worldwide [1]. According to 2022 data from the International World Cancer Research Fund, approximately 20 million people worldwide received new cancer diagnoses, with over one million of them representing hematologic malignancies [2]. In Türkiye, 195,581 individuals were diagnosed with cancer in 2020 [3]. Furthermore, 2024 mortality statistics indicate that one in five deaths resulted from malignant or benign tumors [4]. More than 15,000 new cases of hematologic cancer were reported in Türkiye in 2020, representing approximately 8% of all cancer diagnoses [3].

Cancer has physical and profound psychological impacts on individuals. One of the most prevalent psychological consequences cancer patients experience is fears of cancer recurrence (FCR), commonly observed across all cancer types [5,6]. FCR is defined as fear, worry, or anxiety related to the possibility that the primary cancer may return, a secondary malignancy may develop, or the disease may spread to other parts of the body [7]. Research suggests that features such as preoccupation with death, feelings of loneliness, strong beliefs that the cancer will return, intolerance of uncertainty, intrusive thoughts, daily distressing imagery lasting at least 30 min, and impaired daily functioning characterize clinical FCR levels [6,7,8,9].

Research indicates that FCR is prevalent among cancer survivors, persisting from the time of diagnosis [5,6,10]. Pizzo et al. [11] found that 15.7% of adult survivors of childhood cancer reported high FCR, with 16.6% experiencing clinically significant FCR. Another study reported that 19% of participants experienced severe FCR requiring clinical intervention, with FCR prevalence reaching 80% among leukemia and non-Hodgkin lymphoma patients [5]. Women consistently report higher FCR than men [12,13,14]. In Türkiye, a study of lymphoma patients found that 50.8% experienced high FCR [15]. Although a certain level of FCR may serve an adaptive function for patients (e.g., adherence to medical follow-up, adoption of healthier lifestyle changes), elevated levels can negatively affect quality of life [8]. Given FCR’s significant prevalence and impact across cancer populations, valid and reliable assessment tools are essential for clinical practice and research.

FCR is associated with impaired quality of life and psychosocial adjustment, high emotional distress, and a range of physical symptoms [13,16,17,18,19]. Research has determined that having clinically significant FCR is associated with high anxiety or depression and self-perceived poor health status [11,13,17,19]. Another study revealed that health-care utilization patterns of cancer survivors are linked to FCR. Specifically, frequent health-care utilization in the early stages of treatment has been associated with high FCR levels in later stages. High FCR has been found to be associated with greater health-care utilization during the cancer survivorship process, and this relationship remains statistically significant even when usual health-care utilization is taken into account [20]. These findings demonstrate that FCR represents a psychological burden and can also affect patient quality of life by increasing demands on the health-care system. Research conducted with lymphoma patients in Türkiye found that high FCR levels were positively associated with low levels of quality of life and high levels of anxiety [15].

Hematological malignancies present distinct psychological challenges that influence FCR’s expression and assessment. Unlike solid tumors, these cancers often require prolonged, intensive treatments, including extended hospitalizations, multiple chemotherapy cycles, and continuous monitoring for complications [21,22,23]. These treatment regimens’ chronic nature, coupled with frequent medical interactions, may heighten FCR by serving as persistent reminders of cancer vulnerability. Moreover, the complex and uncertain prognoses associated with hematological cancers can amplify anxiety about disease progression, further intensifying FCR. Patients often face significant physical and psychological burdens because of these demanding treatments, with FCR emerging as a critical concern requiring tailored assessment strategies [24].

The use of validated assessment tools is recommended in both research and clinical settings to enhance FCR management [25]. However, many existing instruments designed to measure individual FCR levels provide limited evidence supporting their conceptual and theoretical foundations, validity, reliability, sensitivity, interpretability, and cross-cultural applicability [24]. In Türkiye, few measurement tools are available to assess FCR among cancer patients, and their validity and reliability for hematologic cancer patients remains unknown. The FCR-4 and FCR-7 scales Humphris et al. [26] recently developed are promising solutions to these assessment challenges because of their several key strengths.

FCR-7 consists of four items assessing the level of concern about cancer recurrence, two items evaluating the extent to which FCR affects patient thoughts and activities, and one item assessing patient behavioral responses. The final item represents a scale reflecting patient experiences regarding FCR severity. FCR-4 includes the first four items of FCR-7 [26]. Subsequent researchers examining the psychometric properties of FCR-4 and FCR-7 have suggested that the sixth item of FCR-7 has inadequate psychometric properties compared to other items of FCR-7 [12,13]. Therefore, FCR-6 represents the version created by removing the sixth item from FCR-7. The construct, convergent, and discriminant validity; known-groups validity; and reliability of FCR4, FCR6, and FCR7 have been examined in different cancer patient samples and different cultures. A general conclusion of these studies is that FCR4, FCR6, and FCR7 represent valid and reliable measurement instruments with a single-factor structure and that women generally have higher FCR levels than men [12,13,17,18,19,27,28].

These scales offer several key advantages for FCR assessment. First, their brief format (4–7 items) makes them highly practical for routine clinical screening and reduces patient burden. This is particularly important for cancer patients who may experience fatigue or distress during lengthy assessments. Second, these scales have demonstrated robust psychometric properties across diverse populations and cancer types. Previous validation studies have consistently reported strong internal consistency, adequate test–retest reliability, and clear single-factor structures across breast, lung, colorectal, and other cancer populations [12,13,17,18,19,27]. Third, these scales have been successfully adapted for cross-cultural use in Brazilian, Spanish, Chinese, and Tamil populations, supporting their applicability in international research. Fourth, FCR scales allow for quick identification of patients requiring psychological intervention through their ability to distinguish clinical FCR levels. This makes them valuable screening tools for oncology settings where rapid assessment is essential.

Despite their established strengths and widespread use, the psychometric properties of FCR-4, FCR-6, and FCR-7 have not been established specifically in hematological cancer patients or validated for use in Turkish health-care contexts. Hematological cancers’ unique characteristics make it essential to ascertain whether these scales maintain their psychometric properties within this specific patient population. Türkiye’s annual burden of more than 15,000 new hematological cancer cases [3] underscores the need for culturally appropriate and validated FCR assessment tools to improve psycho-oncological care in Turkish health-care settings. In this context, examining the psychometric properties of FCR-4, FCR-6, and FCR-7 can help to better understand the concerns of patients experiencing FCR and provide more effective supportive care to them. Similarly, identifying and supporting individuals with FCR who require intervention can enhance psychological well-being, improve health outcomes, increase life expectancy, and elevate overall quality of life [29].

This study focused specifically on adult patients (aged 18 years and older) with hematological malignancies because FCR manifestation and measurement may differ between adult and pediatric populations. This study aimed to adapt FCR-4, FCR-6, and FCR-7 to Turkish and examine their psychometric properties specifically in adult hematological cancer patients receiving treatment at state hospitals in four different provinces of Türkiye. The multi-center approach was chosen to enhance the generalizability of findings across diverse Turkish health-care settings and patient demographics. The specific objectives were to examine FCR scales’ construct validity and factor structure; determine measurement invariance across gender groups; assess convergent and discriminant validity; evaluate known-groups validity across gender; and establish reliability, including both internal consistency and test–retest reliability of the scales in this population.

## 2. Materials and Methods

### 2.1. Participants

We conducted this study with two different samples. In the first sample, we performed construct, convergent, and discriminant validity analyses as well as item analyses of the scales. In the second sample, we conducted test–retest reliability analyses. We selected both samples using convenience sampling. A total of 239 patients participated across the studies. Eligible participants were adults aged 18 years or older with a confirmed diagnosis of hematological cancer. All participants were required to be in remission, receiving treatment at participating hematology clinics, and capable of providing informed consent. Patients were excluded if they had active disease, cognitive impairments that would prevent questionnaire completion, severe psychiatric disorders, or inability to communicate in Turkish.

### 2.2. Measures

Personal Information Form: We developed this form to obtain information about participants’ gender, age, education level, cancer type, cancer duration, and employment status.

FCR-7: We used FCR-7, which Humphris et al. [26] developed at St. Andrews University. The original FCR-7 consists of seven items. The first six items are scored on a 5-point scale (1 = Never, 5 = All the time). The seventh item assesses the extent to which FCR affects the patient’s thoughts and activities and is evaluated using an 11-point Likert-type scale (0 = Not at all, 10 = A great deal). FCR-4 comprises the first four items of FCR-7, whereas FCR-6 consists of all items of FCR-7 except the sixth item [12,13]. Scores range from 6 to 40 for FCR-7, 4 to 20 for FCR-4, and 5 to 35 for FCR-6. Higher scores on all scales indicate greater FCR.

Turkish Adaptation of FCR-7: We followed the International Test Commission’s [30] guidelines for test translation and adaptation in adapting FCR-7 to Turkish. Before beginning the adaptation, validity, and reliability studies of the scale, we obtained all necessary permissions for use and adaptation from the original scale authors. In the second stage, two experts (second and third authors) with native-level language proficiency in English and experience in scale development and adaptation independently translated the scale into Turkish. We then compared these two translation versions, discussed the minor differences identified, and reached consensus on the translation. The first author and a qualified professional translator back-translated the prepared Turkish translation independently into English. When we compared these back-translations, we found no semantic or expressional differences. Following this stage, we submitted the draft Turkish FCR-7 scale for review by an expert panel consisting of mental health professionals, a hematologist, a measurement and evaluation expert, and physicians from different specialties. The experts indicated that all items were grammatically and semantically clear, comprehensible, and appropriate for Turkish culture. Additionally, we conducted a pilot study with 15 cancer patients to evaluate the comprehensibility of the scale by the target population. We asked cancer patients to complete the draft Turkish FCR-7 and highlight any items they found unclear. Qualitative feedback from semi-structured interviews conducted at the end of the pilot study revealed that all participants found the scale items to be clearly worded and conceptually accessible. They reported no challenges with item interpretation or comprehension. Based on these positive feedback responses, we determined that no revision to the scale was necessary. Consequently, we completed the Turkish adaptation process of FCR-7 in accordance with international standards and successfully confirmed the usability of the scale among cancer patients.

Depression Anxiety Stress Scale-21 (DASS-21): We used DASS-21 [31] to determine the level and intensity of depression, anxiety, and stress symptoms in hematologic cancer patients. DASS-21 is a measurement instrument that contains 21 items and three different subscales. These subscales are depression, anxiety, and stress. Each subscale consists of seven items. Şahin et al. [32] conducted validity and reliability analyses of the Turkish version of DASS-21 with an adult sample, whereas Güven et al. [33] conducted these analyses specifically with hematologic cancer patients. Participants rated the extent to which each item applied to them over the past week, using a 4-point Likert scale ranging from 0 (Did not apply to me at all) to 3 (Applied to me very much or most of the time). Scores for each subscale ranged from 0 to 21. Higher scores indicated high levels of depression, anxiety, and stress symptoms, respectively. The Cronbach’s alpha internal consistency coefficient was 0.87 for the depression subscale, 0.83 for the anxiety subscale, and 0.83 for the stress subscale. A sample item from the depression subscale was “I felt down-hearted and blue”; a sample item from the anxiety subscale was “I felt scared without any good reason”; and a sample item from the stress subscale was “I felt that I was rather touchy.”

Satisfaction with Life Scale (SWLS): We used the SWLS developed by Diener et al. [34] to determine life satisfaction levels among participants. Dağlı and Baysal [35] conducted the Turkish adaptation, validity, and reliability analyses of this scale. The scale consisted of five items rated on a seven-point Likert scale ranging from 1 (strongly disagree) to 7 (strongly agree). Total scores ranged from 5 to 35, with higher scores indicating greater life satisfaction. In this study, the internal consistency reliability (Cronbach’s alpha) for the scale was calculated as 0.85. A sample item from the scale was “I am satisfied with my life.”

Single-Item Happiness Scale (SIHS): We administered the SIHS developed by Topkaya et al. [36] to determine happiness levels of cancer patients. Participants responded to the question, “Taken your life as a whole, how would you rate your happiness?” Responses were rated on a 10-point Likert scale ranging from 1 (very unhappy) to 10 (very happy). Scores ranged from 1 to 10, with higher scores indicating greater happiness.

Single-Item General Health Scale (SIGHS): We used SIGHS to determine general self-rated health levels among cancer patients. The scale asked participants to respond to the question, “How would you rate your overall health?” Responses were rated on a 10-point Likert scale ranging from 1 (very poor) to 10 (very good). Scores ranged from 1 to 10, with higher scores indicating better subjective health evaluations. Previous research provided strong evidence of the construct validity of single-item health measures by demonstrating their associations with objective health assessments; various indicators of physical, psychological, and functional health; chronic conditions; and health-related behaviors [37,38,39].

### 2.3. Procedure

We collected data from patients in hematology clinics at four different hospitals between March and June 2025. Before initiating the study, we obtained ethical approval from the Çanakkale University Scientific Research and Publication Ethics Committee. This study was also conducted in accordance with the ethical standards for medical research involving human participants outlined in the World Medical Association’s Declaration of Helsinki. We collected data through face-to-face interviews conducted by researchers with hematology patients. We informed patients that their participation was voluntary, that their information would remain confidential, and that they could withdraw from the study at any time without penalty. All patients voluntarily participated. For the test–retest sample, patients completed the data collection form (Personal Information Form, FCR7) again after a one-month interval. The time required for participants to complete the data collection forms ranged from approximately 5 to 20 min, depending on the specific study.

### 2.4. Statistical Analysis

We conducted all statistical analyses using SPSS 27, Stata 15 [40], and Mplus 7.2 [41]. To identify the factor structure that best represented responses to FCR-7, FCR-6, and FCR-4, we tested five competing models based on prior research and theoretical frameworks [13,17,19,26,27]. Model 1 tested the adequacy of a single common factor to account for the covariance pattern among FCR-7, FCR-6, and FCR-4 items. Model 1 was consistent with the unidimensional factor structure originally suggested by Humphris et al. [26]. Models 2–5 allowed correlated error terms between specific item pairs (Model 2: Items 1–2; Model 3: Items 1–7; Model 4: Items 4–5; Model 5: Items 4–7). We tested Models 1 and 2 only for FCR-4, whereas we tested all models for FCR-7 and FCR-6. Because the scale items had different response options, we standardized the items (*M* = 0, *SD* = 1) before conducting confirmatory factor analysis and reliability analyses.

We examined the fit of data to models through goodness-of-fit indices. In this research, we used goodness-of-fit indices that model complexity and sample size did not affect. These indices are recommended for the maximum likelihood parameter estimates with standard errors and a mean- and variance-adjusted chi-square (χ^2^) test statistic method, which is robust to non-normality [41]. These fit indices included χ^2^, Comparative Fit Index (CFI), Root Mean Square Error of Approximation (RMSEA), Tucker–Lewis Index (TLI), and Standardized Root Mean Square Residual (SRMR). A non-significant χ^2^ value indicated excellent fit to data. However, because sample size considerably affected this value, researchers noted that lower χ^2^ values indicated better model-data fit. CFI and TLI values ≥0.95 represented excellent fit and 0.90–0.94 indicated acceptable fit. RMSEA values ≤0.05 indicated excellent fit and 0.06–0.10 indicated acceptable fit. SRMR values ≤0.05 indicated excellent fit and ≤0.08 indicated good fit [42,43,44,45]. For RMSEA values, unlike other fit indices, 90% confidence intervals could be calculated, allowing us to test whether the calculated RMSEA value differed significantly from 0.05. A *p*-value greater than 0.05 suggested an adequate model-data fit. To examine measurement invariance across genders for the best-fitting model of FCR-7, FCR-4, and FCR-6, we conducted multi-group confirmatory factor analyses. Because of the limited sample size, we evaluated measurement invariance using the χ^2^ difference test for nested models, implemented with the Mplus DIFFTEST command.

We used Pearson’s product–moment correlation analysis to examine the convergent and discriminant validity of FCR scale scores. We also used independent samples *t*-tests to examine the known-groups validity of FCR-7, FCR-4, and FCR-6 scores. We used dependent samples *t*-tests to test whether there were significant changes over a one-month interval in FCR-7, FCR-4, and FCR-6 scores. We used Pearson’s product–moment correlation analysis and intraclass correlation coefficient (ICC; two-way mixed, absolute agreement) to examine the test–retest reliability of FCR-7, FCR-4, and FCR-6 scores. We interpreted correlation coefficients using Cohen’s [46] effect size classification and ICC values using the guidelines that Koo and Li [47] proposed.

We calculated item–total correlation values, Cronbach’s alpha internal consistency coefficients, and McDonald’s omega values based on classical test theory to examine the reliability of FCR-7, FCR-4, and FCR6. Additionally, we used the Graded Response Model to calculate item discrimination parameters of each item, based on item response theory. As a general rule, corrected item–total correlations of 0.30 and above are considered adequate [45,48]. According to Baker and Kim [48], item discrimination between 0.01 and 0.34 is very low, item discrimination between 0.35 and 0.64 is low, item discrimination between 0.65 and 1.34 is moderate, item discrimination between 1.35 and 1.69 is high, and item discrimination 1.7 and above is very high. Reliability coefficients above 0.70 indicate that the scales are suitable for screening and research [49,50,51]. We examined floor and ceiling effects to provide additional evidence for the reliability of FCR-7, FCR-4, and FCR-6 scores. The percentage of respondents with the highest possible total score provided information about ceiling effects, whereas the percentage of respondents with the lowest possible total score provided information about floor effects [52]. We accepted a Type I error rate of *p* < 0.05 for all statistical analyses.

## 3. Results

### 3.1. Sociodemographic Profiles of Cancer Patients

Table 1 presents the sociodemographic and clinical characteristics of hematological cancer patients, including gender, age, education level, marital status, employment status, disease duration, and disease type.

As shown in Table 1, the first sample primarily consisted of males (*n* = 136, 65.1%) ranging from 18 to 88 years, with a mean age of 56.44 years (*SD* = 14.85). Participants in this sample were largely elementary school graduates (*n* = 79, 37.8%), married (*n* = 158, 75.6%), and unemployed (*n* = 144, 68.9%). Non-Hodgkin lymphoma was the most frequent disease type (*n* = 88, 42.1%), followed by multiple myeloma (*n* = 51, 24.4%), Hodgkin lymphoma (*n* = 41, 19.6%), chronic lymphocytic leukemia (*n* = 15, 7.2%), and acute myeloid leukemia (*n* = 14, 6.7%). The mean duration of illness was 50.73 months (*SD* = 44.44), with a range of 7 to 229 months. The second sample also showed a male predominance (*n* = 21, 70%) ranging from 20 to 81 years, with a mean age of 57.87 years (*SD* = 14.85). These participants were mostly high school graduates (*n* = 10, 33.3%), married (*n* = 23, 76.7%), and unemployed (*n* = 23, 76.7%). Multiple myeloma was the predominant diagnosis in the second sample (*n* = 14, 46.7%), and the mean disease duration was 45.57 months (*SD* = 44.76), ranging from 8 to 176 months.

### 3.2. Construct Validity of FCR7, FCR4, and FCR6

Table 2 shows the goodness-of-fit indices for alternative models tested to examine the factor structure of FCR-7, FCR-4, and FCR-6. As shown in Table 1, whereas Model 1, Model 2, Model 3, and Model 5 demonstrated adequate fit with the data, Model 2 demonstrated an excellent fit for FCR-7 and FCR-6. Model 2 also showed an excellent fit with the data for FCR-4. These findings indicated that a single-factor structure, allowing correlated error terms between the first and second items, provided an excellent fit with the data of Turkish hematological cancer patients. We also performed subgroup analyses for employed and unemployed individuals and participants with lymphoma (combining non-Hodgkin lymphoma and Hodgkin lymphoma) and multiple myeloma. Results of these subgroup analyses demonstrated that Model 2 also exhibited good fit to the data for these subgroups. However, these results should be interpreted with caution because of the limited sample sizes used in the analyses (Appendix A). The standardized item factor loadings ranged from 0.51 (Item 6) to 0.90 (Item 3) for FCR-7, from 0.83 (Item 4) to 0.92 (Item 3) for FCR-4, and from 0.57 (Item 6; Item 7 of FCR-7) to 0.90 for FCR-6 (Appendix A). All item factor loadings were statistically significant. The proportion of variance explained by the latent variable of FCR at the item level ranged from 0.26 to 0.81 for FCR-7, 0.69 to 0.85 for FCR-4, and 0.32 to 0.81 for FCR-6 (Appendix A).

Table 3 presents the results of measurement invariance analysis by gender for the single-factor model, allowing correlated error terms between the first and second items (Model 2). As shown in Table 2, the χ^2^ difference tests for nested models were not significant for metric, scalar, residual variance levels and residual invariance levels for FCR-7, FCR-4, and FCR-6. These findings indicated that Model 2 exhibited strict measurement invariance across gender for FCR-7, FCR-4, and FCR-6.

### 3.3. Convergent and Discriminant Validity of FCR-7, FCR-4, and FCR-6

Table 4 shows the mean and standard deviation values of the variables and the results of the Pearson product–moment correlation coefficient analysis conducted to determine the strength and direction of relationships among the variables. As shown in Table 4, FCR-7, FCR-4, and FCR-6 scores were moderately positively correlated with depression, anxiety, and stress. In contrast, FCR-7, FCR-4, and FCR-6 scores were weakly negatively correlated with life satisfaction scores and moderately negatively correlated with happiness scores. Finally, FCR-7 and FCR-6 scores were moderately negatively correlated, while FCR-4 scores were weakly negatively correlated, with subjective health scores.

### 3.4. Known-Groups Validity of FCR-7, FCR-4, and FCR-6

Table 5 shows the results of the independent samples *t*-tests conducted to examine the known-groups validity of FCR-7, FCR-4, and FCR-6 scores.

As shown in Table 5, the independent samples *t*-tests revealed significant gender differences for FCR-7 (*t* [207] = 3.26, *p* < 0.01, *d* = 0.47), FCR-4 (*t* [207] = 2.89, *p* < 0.01, *d* = 0.42), and FCR-6 (*t* [207] = 3.17, *p* < 0.01, *d* = 0.46) scores. These observed gender differences in FCR-7, FCR-4, and FCR-6 scores demonstrated small effect sizes. As shown in Table 6, female cancer patients scored significantly higher than male cancer patients on FCR-7, FCR-4, and FCR-6.

### 3.5. Reliability of FCR-7, FCR-4, and FCR-6

Table 6 shows the results of dependent samples *t*-tests conducted to determine whether there were significant changes over time in FCR-7, FCR-4, and FCR-6 scores.

As shown in Table 6, the dependent samples *t*-tests revealed no significant differences between the mean scores of FCR-7, FCR-4, and FCR-6 administered to cancer patients one month apart. These results indicated the temporal stability of the scale scores over a one-month period.

Table 7 shows the correlation coefficients and ICCs for FCR-7, FCR-4, and FCR-6 test–retest reliability over a one-month interval.

As shown in Table 7, the Pearson correlation coefficients for FCR-7, FCR-4, and FCR-6 scores were 0.87, 0.85, and 0.86, respectively, indicating very high correlations between the first and second administrations. The ICC analyses also indicated good reliability for FCR-7, FCR-4, and FCR-6.

Table 8 displays the item–total correlations, Cronbach’s alpha coefficients, McDonald’s omega values, item discrimination parameters based on the graded response model, and floor and ceiling effect rates for FCR-7, FCR-4, and FCR-6.

As shown in Table 8, FCR-7 item–total correlation values ranged from 0.49 to 0.84, FCR-4 item–total correlation values ranged from 0.79 to 0.88, and FCR-6 item–total correlation values ranged from 0.54 to 0.85. Cronbach’s alpha coefficients for internal consistency were 0.91 for FCR-7, 0.93 for FCR-4, and 0.92 for FCR-6. McDonald’s omega coefficients were 0.92 for FCR-7 and FCR-6, and 0.93 for FCR-4. Based on the graded response model, item discrimination values (a) for FCR-7 showed moderate discrimination for Items 6 and 7, while the remaining items showed very high discrimination. All items in FCR-4 demonstrated very high discrimination, while the first five items in FCR-6 demonstrated very high discrimination, and the sixth item (seventh item of FCR-7) demonstrated moderate item discrimination. Across all scales, Item 2 had the highest item discrimination value. The percentage of respondents with the highest possible total score (ceiling effect) was 1.4% for FCR-7, 9.8% for FCR-4, and 5.3% for FCR-6, whereas the percentage with the lowest possible total score (floor effect) was 3.8% for FCR-7, 4.8% for FCR-4, and 1.9% for FCR-6.

## 4. Discussion

This study examined the validity and reliability of three FCR scales (FCR-7, FCR-4, and FCR-6) in hematological cancer patients. To assess construct validity, we tested alternative models and found that the single-factor structure, which allowed correlated error terms between the first and second items, demonstrated superior goodness-of-fit indices compared to other single-factor models and demonstrated excellent fit with the data for all three scales. The model parameter estimates, including item factor loadings, standard errors, and the proportion of variance explained, were adequate [43,44,53]. These findings are consistent with the structure that the scale developers [26] proposed and align with previous research demonstrating a single-factor structure for FCR scales in various samples [12,13,17,18,19,27,28]. For example, Bergerot et al. [27] found that FCR-7 and FCR-4 had a single-factor structure in their research with Brazilian cancer patients. Díaz-Periánez et al. [13] also reported that FCR-6 had a single-factor structure in their research with Spanish cancer patients.

Although researchers consistently support a single-factor structure for FCR scales, they frequently add correlated error terms between specific items [19,27]. In this study, we correlated the error terms of the first and second items because “being afraid” and “being worried/anxious” about cancer recurrence represent highly overlapping emotional responses that participants may interpret almost identically. Thus, their error terms are expected to correlate beyond what the latent FCR factor explains. Future research should test models that allow these error terms to be correlated when examining FCR scale construct validity using confirmatory factor analysis.

This study also found that FCR-7, FCR-4, and FCR-6 demonstrated configural, metric, scalar, and strict measurement invariance across gender. The configural invariance of FCR-7, FCR-4, and FCR-6 indicates that women and men conceptualize FCR in the same way or that the same item–factor structure exists across groups. Practically, these findings imply that researchers can use the same scale structure for both male and female patients when screening FCR. The metric invariance of FCR-7, FCR-4, and FCR-6 also suggests that the strength of the relationship between FCR-7, FCR-4, and FCR-6 items and the latent FCR factor is equivalent between women and men, or more simply, that the same latent factor is measured in each group. Practically, these findings denote that if a man and a woman have the same underlying level of FCR, they interpret and respond to each item in a comparable way. Therefore, researchers can confidently compare latent associations between FCR and other variables (e.g., depression, quality of life) across genders. The scalar invariance of FCR-7, FCR-4, and FCR-6 reveals that individuals with the same level of latent FCR will have equivalent observed scores, regardless of gender. This means that any observed differences in FCR scale scores between men and women reflect true differences in underlying FCR levels rather than measurement bias. For example, if women score higher than men on average, this difference reflects real higher FCR rather than gender-based measurement artifacts. Finally, the strict invariance of FCR-7, FCR-4, and FCR-6 indicates that the amount of item variance not accounted for by latent FCR is similar across groups and that there are no significant differences in error covariance between the first and second items across groups. This demonstrates that the scales measure FCR with equivalent precision across genders. Because the strict invariance of FCR-7, FCR-4, and FCR-6 indicates that group differences by gender are due only to true differences in means, researchers can validly compare observed and latent mean differences, variances, and covariances by gender [43,44,53].

The findings of the convergent and discriminant validity analysis indicate that FCR7, FCR4, and FCR6 scale scores are negatively associated with variables measuring psychological well-being (happiness, general health evaluations, life satisfaction) at low to moderate magnitudes and moderately positively associated with variables measuring similar qualities (depression, anxiety, and stress). The present findings are consistent with previous research that has demonstrated FCR scales’ convergent and discriminant validity [12,13,17,18,19]. For example, Yang et al. [17] and Lee et al. [19] reported that FCR-7 scores were positively associated with depression and anxiety scores among cancer patients. Díaz-Periánez et al. [13] found that both FCR-7 and FCR-6 scores were negatively associated with subjective health ratings. Similarly, Nandakumar et al. [18] found that FCR-7 scores were negatively related to quality-of-life scores. Taken together, the moderate correlations between FCR scale scores and general psychological distress indicators show that FCR differentiates measurably from these broader concepts. Furthermore, the negative correlations between FCR scale scores and subjective well-being indicators are expected but not strong, indicating that FCR represents a conceptually separate construct from psychological well-being indicators. These correlation patterns provide strong evidence that the scales can appropriately distinguish both related and distinct constructs as required.

Known-groups validity refers to a type of construct validity that evaluates an instrument’s ability to detect significant differences between groups that are theoretically or empirically expected to differ. In this study, results revealed that women scored significantly higher than men on all versions of the FCR scales (FCR-7, FCR-4, and FCR-6). These findings support those of previous research. For example, a meta-analysis conducted by Pang and Humphris [14] reported that women tend to experience higher levels of FCR than men. Similarly, studies using different versions of FCR scales [12,13] have consistently shown that women experience a higher FCR compared to men. The present findings support the FCR scales’ known-groups validity by confirming their capacity to reliably distinguish gender-based differences in FCR.

The results of the test–retest reliability, item analysis, internal consistency, and floor and ceiling effect analyses support the reliability of the FCR scales. The test–retest reliability analyses demonstrated that the FCR scales had good temporal stability over a one-month interval and produced similar and relatively stable scores over time [46,47]. These findings imply that researchers can use the FCR scales to examine differences in FCR scores between and within groups over time, and to test the effectiveness of treatments and interventions. Item-level analyses based on classical test theory further showed that item–total correlations for FCR-7 ranged from moderate to high, whereas all items in FCR-4 and FCR-6 showed high item–total correlations. These results indicate that higher item scores are consistently associated with higher overall levels of FCR and that each item is a strong indicator of the underlying FCR construct [54]. The item response theory analysis of the FCR scales also showed high discrimination for most items, with the exception of the sixth and seventh items of FCR-7 and the sixth item of FCR-6, which had moderate discrimination. These findings are consistent with previous item response theory analyses of FCR-7 by Humphris et al. [26].

Similarly, McDonald’s omega and Cronbach’s alpha reliability values indicate high internal consistency among items across all FCR scales. These findings demonstrate that FCR scales can be used for screening, clinical assessment and decision-making, and research purposes. These findings support prior studies that also reported high reliability for the FCR scales [12,13,17,18,19,26,27]. Finally, floor and ceiling effects analyses showed that participants with extreme scores comprised ≤15% across all FCR scales, indicating absence of problematic effects and providing additional evidence for scale reliability, content validity, and responsiveness [52].

### 4.1. Practical Implications

The findings of this study demonstrate that the FCR scales (FCR-7, FCR-6, and FCR-4) are reliable and valid instruments for assessing FCR in patients with hematological cancers. Clinicians and researchers can confidently use different versions of these scales to evaluate FCR levels among hematologic cancer patients, depending on their specific needs or research purposes. The measurement invariance of the FCR scales across gender indicates that these scales can be appropriately used to compare FCR levels between male and female patients. This enables gender-based comparisons and intervention planning. The known-groups validity of the scales suggests that they are capable of distinguishing between individuals at varying levels of risk, thereby supporting their use in identifying high-risk patients for targeted support, early screening, and risk stratification in oncology clinics. The high test–retest reliability of the scales also indicates that these scales are suitable for use in longitudinal studies and evaluations of treatment or intervention outcomes. Finally, the absence of floor and ceiling effects in the FCR scales denotes that these instruments can assess the full spectrum of FCR, from mild to severe symptoms. This characteristic makes them suitable for diverse patient populations across all disease stages. Overall, the strong psychometric characteristics of FCR scales make them valuable tools for routine psycho-oncological assessments. These scales can guide personalized intervention strategies and support longitudinal research to improve patient quality of life. For example, health-care providers can identify patients requiring mental health specialist referrals or evaluate survivorship program effectiveness.

### 4.2. Limitations

This study has several limitations. First, its scope is limited to data collected from only four hospitals (Çorum, İzmir, Manisa, and Çanakkale). Therefore, the generalizability of its findings to all hematological cancer patients in Türkiye is restricted. The external validity of our findings is also limited because the hematological cancer patients we examined represent only a small minority of the global cancer patient population. It is possible that the interpretation and expression of FCR symptoms could differ significantly in various cancer patient groups. Second, because this study used self-report data collection tools, the results are subject to limitations, such as social desirability bias or recall bias. Additionally, participants’ current emotional state, response styles (e.g., acquiescence or extreme responding), or lack of insight into their own psychological processes may have influenced self-report measures. These factors can lead to inaccuracies or distortions in the reported data and may limit the validity of the findings.

Third, this study did not account for other external factors (e.g., socioeconomic status, comorbid physical illnesses, level of social support, or recent stressful life events) that could influence the symptom intensity of FCR and lead to systematic changes in item responses that are not attributable to the latent construct of FCR. Fourth, although subgroup analyses confirmed the validity and reliability of FCR-7, FCR-4, and FCR-6 scales for employed and unemployed participants, as well as for participants with multiple myeloma and lymphoma, the small sample sizes in these subgroups may limit these findings’ generalizability. Unemployed patients and those with specific cancer types may experience unique psychosocial stressors (e.g., financial instability or disease-specific challenges) that could influence FCR manifestation. Finally, although this research examined the structural, convergent, discriminant validity, and reliability of the FCR scales, it did not investigate their predictive validity. Therefore, future studies could conduct predictive validity analyses to determine the extent to which the FCR scales can predict future health behaviors, quality of life, psychosocial adjustment, and treatment adherence. Consequently, future studies should test the predictive validity of these scales, employ larger and more diverse samples, conduct comparative analyses across different cancer types, and use longitudinal research designs to examine changes in FCR over time.

## 5. Conclusions

This study examined FCR scales’ validity and reliability. The findings indicated that the scales possess a unidimensional structure. Convergent and discriminant validity analyses demonstrated that the scales correlate with subjective well-being indicators (happiness, life satisfaction, subjective health evaluations) and psychological distress indicators (depression, anxiety, stress) in expected directions. Further, the known-groups validity analysis revealed that the scales effectively distinguished between groups and replicated the finding consistently established in previous research that women report significantly higher levels of FCR compared to men. The reliability analyses further demonstrated that all FCR scales are reliable instruments for use among patients with hematological cancers. Overall, the results provide strong psychometric support for the use of FCR-7, FCR-4, and FCR-6 as valid and reliable tools for assessing FCR in Turkish patients with hematological cancers. These scales can help clinicians assess FCR levels in patients and design appropriate treatment strategies.

## Figures and Tables

**Table 1 medicina-61-01628-t001:** Sociodemographic characteristics of study samples.

Variables	First Sample	Second Sample
Sex, *n* (%)		
Female	73 (34.9)	9 (30)
Male	136 (65.1)	21 (70)
Age		
*M* [Min, Max.]	56.44 [18, 88]	57.87 [20, 81]
Education level, *n* (%)		
Primary school	79 (37.8)	8 (26.7)
Secondary school	32 (15.3)	2 (6.7)
High school	45 (21.5)	10 (33.3)
Associate degree	3 (1.4)	1 (3.3)
Bachelor’s degree or above	50 (24)	9 (30)
Marital status, *n* (%)		
Single	40 (19.1)	23 (76.7)
Married	158 (75.6)	7 (23.3)
Missing	11 (5.3)	
Employment status, *n* (%)		
Employed	65 (31.1)	7 (23.3)
Unemployed	144 (68.9)	23 (76.7)
Disease duration		
*M* [Min, Max.]	50.73 [7, 229]	45.57 [8, 176]
Disease type, *n* (%)		
Chronic lymphocytic leukemia	15 (7.2)	3 (10)
Multiple myeloma	51 (21.4)	14 (46.7)
Non-Hodgkin’s lymphoma	88 (42.1)	7 (23.3)
Hodgkin’s lymphoma	41 (19.6)	3 (10)
Acute myeloid leukemia	14 (6.7)	3 (10)

Note. Total sample sizes were *N*_1_ = 209 and *N*_2_ = 30 for the first and second samples, respectively.

**Table 2 medicina-61-01628-t002:** Goodness-of-fit indices for alternative models of FCR-7, FCR-4, and FCR-6.

Models	χ^2^	*df*	*p*	CFI	TLI	RMSEA	RMSEA *p*	LB	UB	SRMR
FCR7										
Model 1	41.799	14	0.001 ***	0.961	0.941	0.097	0.012 *	0.064	0.132	0.036
Model 2	26.752	13	0.013 *	0.981	0.969	0.071	0.164	0.031	0.110	0.028
Model 3	36.839	13	0.001 ***	0.966	0.946	0.094	0.022 *	0.059	0.130	0.033
Model 4	35.233	13	0.001 ***	0.969	0.949	0.090	0.032 *	0.055	0.127	0.037
Model 5	27.342	13	0.011 *	0.980	0.967	0.073	0.148	0.033	0.148	0.028
FCR4										
Model 1	8.182	2	0.017 *	0.991	0.972	0.122	0.062	0.044	0.213	0.015
Model 2	0.815	1	0.367	1.000	1.001	0.000	0.481	0.000	0.176	0.004
FCR6										
Model 1	34.012	9	0.001 ***	0.963	0.938	0.115	0.005 **	0.076	0.158	0.036
Model 2	20.735	8	0.008 **	0.981	0.965	0.087	0.082	0.042	0.134	0.028
Model 3	28.632	8	0.001 ***	0.969	0.943	0.111	0.011 *	0.069	0.156	0.032
Model 4	26.420	8	0.001 ***	0.973	0.949	0.105	0.020 *	0.062	0.151	0.036
Model 5	21.705	8	0.006 **	0.977	0.957	0.091	0.065	0.046	0.137	0.028

Note. FCR7 = Fears of Cancer Recurrence-7 Item Version; FCR4 = Fears of Cancer Recurrence-4 Item Version; FCR6 = Fears of Cancer Recurrence-6 Item Version; *p* < 0.05 *, *p* < 0.01 **, *p* < 0.001 ***.

**Table 3 medicina-61-01628-t003:** Results of measurement invariance analysis for FCR-7, FCR-4, and FCR-6 across genders.

Invariance Level	NFP	χ^2^	*df*	*p*	CFI	TLI	RMSEA	Comparison Model	Δχ^2^	*p*
Model 2										
FCR7										
1. Configural	44	36.471	26	0.083	0.983	0.973	0.062	-	-	-
2. Metric	38	41.589	32	0.119	0.985	0.980	0.054	2 vs. 1	4.086	0.665
3. Scalar	32	44.117	38	0.229	0.990	0.989	0.039	3 vs. 2	0.523	0.998
4. Residual variance	25	49.713	45	0.291	0.993	0.993	0.032	4 vs. 3	6.503	0.482
5. Residual covariance	24	49.492	46	0.336	0.994	0.995	0.027	5 vs. 4	1.185	0.276
FCR4										
1. Configural	26	0.836	2	0.658	1.00	1.012	0.000	-	-	-
2. Metric	23	3.193	5	0.670	1.00	1.007	0.000	2 vs. 1	2.825	0.420
3. Scalar	20	3.513	8	0.898	1.00	1.012	0.000	3 vs. 2	0.00	1.00
4. Residual variance	16	6.811	12	0.870	1.00	1.009	0.000	4 vs. 3	2.584	0.630
5. Residual covariance	15	6.901	13	0.907	1.00	1.010	0.000	5 vs. 4	0.073	0.787
FCR6										
1. Configural	38	18.702	16	0.284	0.995	0.991	0.040	-	-	-
2. Metric	33	23.197	21	0.334	0.996	0.994	0.032	2 vs. 1	4.046	0.543
3. Scalar	28	25.441	26	0.494	1.00	1.001	0.000	3 vs. 2	0.00	1.00
4. Residual variance	22	30.757	32	0.529	1.00	1.002	0.000	4 vs. 3	5.296	0.507
5. Residual covariance	21	30.782	33	0.578	1.00	1.004	0.000	5 vs. 4	1.167	0.280

Note. FCR7 = Fears of Cancer Recurrence-7 Item Version; FCR4 = Fears of Cancer Recurrence-4 Item Version; FCR6 = Fears of Cancer Recurrence-6 Item Version; NFP = Number of free parameters.

**Table 4 medicina-61-01628-t004:** Results of Pearson correlation analysis.

Scales	*M*	*SD*	FCR7	FCR4	FCR6
1. FCR7	21.24	8.38			
2. FCR4	11.14	4.33	0.92	-	
3. FCR6	18.32	7.67	0.99	0.92	-
Convergent validity					
4. Depression	3.22	3.55	0.34	0.33	0.36
5. Anxiety	3.81	3.51	0.41	0.37	0.42
6. Stress	4.68	3.75	0.46	0.45	0.46
Discriminant validity					
7. Life satisfaction	25.91	5.66	−0.25	−0.27	−0.27
8. Subjective health	7.65	1.98	−0.33	−0.29	−0.34
9. Happiness	8.12	2.00	−0.32	−0.30	−0.33

Note. *N* = 209; FCR7 = Fears of Cancer Recurrence-7 Item Version; FCR4 = Fears of Cancer Recurrence-4 Item Version; FCR6 = Fears of Cancer Recurrence-6 Item Version. All correlation coefficients were statistically significant at least at the *p* < 0.001 level.

**Table 5 medicina-61-01628-t005:** Results of independent samples *t*-test for FCR-7, FCR-4, and FCR-6 scores.

Gender	*n*	*M*	*SD*	*df*	*t*	*p*	*d*
FCR7							
Female	73	23.77	8.00	207	3.26	0.001 **	0.47
Male	136	19.89	8.29				
FCR4							
Female	73	12.30	4.12	207	2.89	0.004 **	0.42
Male	136	10.51	4.33				
FCR6							
Female	73	20.56	7.28	207	3.17	0.002 **	0.46
Male	136	17.11	7.63				

Note. FCR7 = Fears of Cancer Recurrence-7 Item Version; FCR4 = Fears of Cancer Recurrence-4 Item Version; FCR6 = Fears of Cancer Recurrence-6 Item Version; *p* < 0.01 **.

**Table 6 medicina-61-01628-t006:** Results of dependent samples *t*-test for FCR-7, FCR-4, and FCR-6 scores.

Administration Time	*n*	*M*	*SD*	*df*	*t*	*p*	*d*
FCR7							
First administration	30	19.63	6.82	29	1.45	0.157	0.27
Second administration	30	18.67	7.24				
FCR4							
First administration	30	11.20	3.92	29	1.73	0.094	0.32
Second administration	30	10.50	4.18				
FCR6							
First administration	30	16.80	6.26	29	1.30	0.205	0.24
Second administration	30	15.97	6.76				

Note. FCR7 = Fears of Cancer Recurrence-7 Item Version; FCR4 = Fears of Cancer Recurrence-4 Item Version; FCR6 = Fears of Cancer Recurrence-6 Item Version.

**Table 7 medicina-61-01628-t007:** Pearson correlation coefficients and ICCs for test–retest reliability of FCR7, FCR4, and FCR6.

Administration Time	*n*	*r*	ICC	ICC % 95 CI	
				Lower Bound	Upper Bound
FCR7					
First administration	30	0.87	0.86	0.73	0.93
Second administration	30				
FCR4					
First administration	30	0.85	0.84	0.69	0.92
Second administration	30				
FCR6					
First administration	30	0.86	0.85	0.71	0.93
Second administration	30				

Note. FCR7 = Fears of Cancer Recurrence-7 Item Version; FCR4 = Fears of Cancer Recurrence-4 Item Version; FCR6 = Fears of Cancer Recurrence-6 Item Version. All correlation coefficients were statistically significant at least at the *p* < 0.001 level.

**Table 8 medicina-61-01628-t008:** Results of FCR-7, FCR-4, and FCR-6 reliability analysis.

Item Number	*M*	*SD*	FCR7	FCR7	FCR4	FCR4	FCR6	FCR6
			*r*	a	*r*	a	*r*	a
Item 1	2.96	1.25	0.79	4.18	0.83	4.62	0.80	4.26
Item 2	2.84	1.18	0.84	5.70	0.88	7.04	0.85	5.83
Item 3	2.82	1.14	0.84	4.80	0.86	4.63	0.84	4.81
Item 4	2.52	1.19	0.83	3.78	0.79	3.14	0.84	3.72
Item 5	2.67	1.13	0.80	3.54			0.79	3.72
Item 6	2.93	1.35	0.49	1.29				
Item 7	4.50	3.41	0.54	1.30			0.54	1.30
Mean *r*			0.59		0.77		0.66	
α			0.91		0.93		0.92	
ω			0.92		0.93		0.92	
HPS (%)			1.4		9.1		5.3	
LPS (%)			3.8		4.8		1.9	

Note. FCR-7 = Fears of Cancer Recurrence-7 Item Version; FCR-4 = Fears of Cancer Recurrence-4 Item Version; FCR-6 = Fears of Cancer Recurrence-6 Item Version; *r* = Item–total correlation value; a = Item response theory-based item discrimination value; HPS = Highest possible score; LPS = Lowest possible score. All correlation values were significant at least at *p* < 0.001.

## Data Availability

Data underlying this article are available from the Open Science Framework (osf.io/krhye).

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
