# Peer review of "Psychometric Properties of Fears of Cancer Recurrence Scales in Turkish Hematologic Cancer Patients"

_medicina, 2025, doi:10.3390/medicina61091628_

Round 1

Reviewer 1 Report

Comments and Suggestions for Authors

I commend the authors to have thought out every aspect of the study. I appreciate having the translations done by 2 different authors. 

It would be nice if there can be a sub group analysis of patients who are unemployed in your study as it is a big number and see if the scales are valid then too. I see the limitation in discussion section

Similarly, the quality of life of patients with different malignancies is different such as those with hodgkins have complete cures while multiple myeloma doesnt have a complete cure- so the FCR  will be different for them. Please try to do a sub group analysis of the disease as well an make a table for that. 

I do think table 1&2 are excessive and not required. 

Author Response

Comments of Reviewer 1

Response

Comment#1. I commend the authors to have thought out every aspect of the study. I appreciate having the translations done by 2 different authors

Thanks for this positive comment. There is nothing to change for this comment.

Comment#2. It would be nice if there can be a sub group analysis of patients who are unemployed in your study as it is a big number and see if the scales are valid then too. I see the limitation in discussion section

Thanks for this helpful comment. We added following to Limitations section.

We conducted subgroup analyses as reviewer requested. Model 2 also excellent fit with the data for employed and unemployed. We report this findings the supplementary material Table S1 and Table S2. We also added following to results section.

Results of these subgroup analyses demonstrated that Model 2 also exhibited good fit to the data for these subgroups. However, these results should be interpreted with cau-tion due to the limited sample sizes in these analyses ((Supplementary Material Table S1-Table S4).

We also added following to Limitation section.

Fourth, although subgroup analyses confirmed the validity and reliability of the FCR-7, FCR-4, and FCR-6 scales for employed and unemployed participants, as well as for participants with multiple myeloma and lymphoma, the small sample sizes in these subgroups may limit the generalizability of these findings. Unemployed patients and those with specific cancer types may experience unique psychosocial stressors (e.g., fi-nancial instability or disease-specific challenges) that could influence FCR manifesta-tion.

Comment#3. Similarly, the quality of life of patients with different malignancies is different such as those with hodgkins have complete cures while multiple myeloma doesnt have a complete cure- so the FCR  will be different for them. Please try to do a sub group analysis of the disease as well an make a table for that.

Thanks for this helpful comment. We added following to Limitations section.

We conducted subgroup analyses as reviewer requested. Model 2 also excellent fit with the data for multiple myeloma and lymphoma. We report this findings the supplementary material Table S3 and Table S4.

We also added following to results section.

Results of these subgroup analyses demonstrated that Model 2 also exhibited good fit to the data for these subgroups. However, these results should be interpreted with cau-tion due to the limited sample sizes in these analyses ((Supplementary Material Table S1-Table S4).

We also added following to Limitation section.

Fourth, although subgroup analyses confirmed the validity and reliability of the FCR-7, FCR-4, and FCR-6 scales for employed and unemployed participants, as well as for participants with multiple myeloma and lymphoma, the small sample sizes in these subgroups may limit the generalizability of these findings. Unemployed patients and those with specific cancer types may experience unique psychosocial stressors (e.g., fi-nancial instability or disease-specific challenges) that could influence FCR manifesta-tion.

Comment#4. I do think table 1&2 are excessive and not required.

We removed Table 2 from the manuscript. Because Table 1 is necessary, it is retained.

The standardized item factor loadings ranged from 0.51 (Item 6) to 0.90 (Item 3) for FCR-7, from 0.83 (Item 4) to 0.92 (Item 3) for FCR-4, and from 0.57 (Item 6; Item 7 of FCR-7) to 0.90 for FCR-6 (Supplementary Material Table S5). All item factor loadings were statistically significant. The proportion of variance explained by the la-tent variable of FCR at the item level ranged from 0.26 to 0.81 for the FCR-7, 0.69 to 0.85 for the FCR-4, and 0.32 to 0.81 for the FCR-6 (Supplementary Material Table S5).

Comment #5.  The English is fine and does not require any improvement.

Thanks for this positive comment. There is nothing to change for this comment.

Reviewer 2 Report

Comments and Suggestions for Authors

Abstract: I would describe the findings with numerical values and not discuss the results here. In the results, clinical assessments and screening are the same. I would rephrase this. 

Introduction:

  • Needs major editing to focus on the topic of the study. Would recommend talking about the main strengths of the FCR scales. The introduction and discussion are two different things. 
    - Limitations: Needs stronger references for the claims on line 109
  • Aims can include the population and the center in which the study was conducted. Were adults the only focus here? 

Methods:

  • The description of the study population is more suited for the results portion.
  • The statistical analysis can be split into study outcomes and the analytical methods 

Discussion:

  • Maintain consistency with the citation style
  • Has to be cut down in size. Para 2 describes the error terms, but gives a detailed report. Has to be concise to maintain the readers' interest. 
  • A repetition of results is seen in the subsequent paras. Would rephrase these areas

Practical applications are a better fit for the conclusion 

Author Response

Comments of Reviewer 2

Response

Comment#1. Abstract: I would describe the findings with numerical values and not discuss the results here. In the results, clinical assessments and screening are the same. I would rephrase this.

Thanks for this helpful comment. We revised the abstract as below. 

Fear of cancer recurrence (FCR) represents one of the most common psychological problems in cancer patients. Therefore, there is a need for valid and reliable measure-ment tools to assess FCR in this population. The purpose of this study was to examine the psychometric properties of the FCR Scales (Fears of Cancer Recurrence-7 Item Version [FCR-7]; Fears of Cancer Recurrence-4 Item Version [FCR-4]; Fears of Cancer Re-currence-6 Item Version [FCR-6]) in Turkish hematological cancer patients. The study sample consisted of 242 hematological cancer patients undergoing treatment at four different state hospitals in Türkiye. Confirmatory factor analysis results showed that all three scales have a single-factor structure (FCR-7: CFI = 0.981, TLI = 0.969, RMSEA = 0.071, SRMR = 0.028; FCR-4: CFI = 1.000, TLI = 1.001, RMSEA = 0.000, SRMR = 0.004; FCR-6: CFI = 0.981, TLI = 0.965, RMSEA = 0.087, SRMR = 0.028). The model allowing for correlated error terms between the first and second items provided the best fit. Re-search findings also indicated that the scales possess strict measurement invariance across gender. Convergent and discriminant validity analyses also demonstrated expected associations between FCR scale scores and psychological well-being (r = -0.25 to -0.34) and psychological distress (r = 0.33 to 0.46) variables. The known-groups validity analysis indicated that the scales are effective at differentiating between groups and that they replicated the well-established finding from previous research that women report significantly higher levels of FCR than men (d = 0.42-0.47). Reliability analyses indicated that all three versions of the FCR scales had high internal consistency (Cronbach's α = 0.91-0.93) and strong test–retest reliability (r = 0.85-0.87). Overall, the findings indicate that FCR-7, FCR-4, and FCR-6 are psychometrically sound, valid, and reliable instruments for assessing FCR in Turkish hematological cancer patients. These instruments can be used in clinical practice and research studies, as well as for evaluating intervention effectiveness in this population.

Comment#2. Needs major editing to focus on the topic of the study. Would recommend talking about the main strengths of the FCR scales. The introduction and discussion are two different things. Limitations: Needs stronger references for the claims on line 109

Thanks for this helpful comment. We added following to the Introduction section.

Hematological malignancies present distinct psychological challenges that influence the expression and assessment of FCR. Unlike solid tumors, these cancers often require prolonged, intensive treatments, including extended hospitalizations, multiple chemotherapy cycles, and continuous monitoring for complications. The chronic nature of these treatment regimens, coupled with frequent medical interactions, may heighten FCR by serving as persistent reminders of cancer vulnerability. Moreover, the complex and uncertain prognoses associated with hematological cancers can amplify anxiety about disease progression, further intensifying FCR. Patients often face significant physical and psychological burdens due to these demanding treatments, with FCR emerging as a critical concern requiring tailored assessment strategies.

The use of validated assessment tools is recommended in both research and clinical settings to enhance the management of FCR. However, many existing instruments designed to measure individual FCR levels exhibit limited evidence supporting their conceptual and theoretical foundations, validity, reliability, sensitivity, interpretability, and cross-cultural applicability. In Türkiye, few measurement tools are available to assess FCR among cancer patients, and their validity and reliability for hematologic cancer patients remains unknown.  The FCR-4 and FCR-7 scales recently developed by Humphris et al., represent promising solutions to these assessment challenges due to several key strengths.

FCR-7 consists of four items assessing the level of concern about cancer recurrence, two items evaluating the extent to which FCR affects patient thoughts and activities, and one item assessing patient behavioral responses. The final item represents a scale reflecting patient experiences regarding FCR severity. FCR-4 includes the first four items of FCR-7 [23]. Subsequent researchers examining the psychometric proper-ties of FCR-4 and FCR-7 have suggested that the sixth item of FCR-7 has inadequate psychometric properties compared to other items of FCR-7. Therefore, FCR-6 represents the version created by removing the sixth item from FCR-7. The construct, convergent and discriminant validity, known-groups validity, and reliability of FCR4, FCR6, and FCR7 have been examined in different cancer patient samples and different cultures. A general conclusion of these studies is that FCR4, FCR6, and FCR7 represent valid and reliable measurement instruments with a single-factor structure and that women generally have higher FCR levels than men.

These scales offer several key advantages for FCR assessment. First, their brief format (4-7 items) makes them highly practical for routine clinical screening and re-duces patient burden, particularly important for cancer patients who may experience fatigue or distress during lengthy assessments. Second, these scales have demonstrated robust psychometric properties across diverse populations and cancer types. Previous validation studies have consistently reported strong internal consistency, adequate test-retest reliability, and clear single-factor structures across breast, lung, colorectal, and other cancer populations. Third, these scales have been successfully adapted for cross-cultural use in Brazilian, Spanish, Chinese, and Tamil populations, supporting their applicability in international research. Fourth, the FCR scales provide quick identification of patients requiring psychological intervention through their ability to distinguish clinical levels of FCR, making them valuable screening tools for oncology settings where rapid assessment is essential.

Despite their established strengths and widespread use, the psychometric proper-ties of FCR-4, FCR-6, and FCR-7 have not been established specifically in hematological cancer patients or validated for use in Turkish healthcare contexts. The unique char-acteristics of hematological cancers make it essential to ascertain whether these scales maintain their psychometric properties within this specific patient population. Türkiye's annual burden of more than 15,000 new hematological cancer cases [3] under-scores the need for culturally appropriate and validated FCR assessment tools to im-prove psycho-oncological care in Turkish healthcare settings. In this context, examining the psychometric properties of FCR-4, FCR-6, and FCR-7 can help to better under-stand the concerns of patients experiencing FCR and provide more effective support-ive care to these patients. Similarly, identifying and supporting individuals with FCR who require intervention can enhance their psychological well-being, improve health outcomes, increase life expectancy, and elevate their overall quality of life [29].

We also removed following from Introduction

Advanced FCR is conceptualized as a multidimensional construct, characterized by four core features: (a) high levels of preoccupation, (b) intense worry, (c) persistent anxiety, and (d) heightened sensitivity to physical symptoms [9]. FCR is shaped by cognitive, emotional, and metacognitive processes, elucidated through several theoretical frameworks. Relational Frame Theory (RFT) underscores existential concerns, emphasizing psychological flexibility and acceptance as critical factors. The Self-Regulatory Executive Function (S-REF) model highlights worry, rumination, and maladaptive coping strategies as key contributors to sustained anxiety. Meanwhile, the Common-Sense Model (CSM) centers on illness representations and threat appraisals as pivotal to FCR. Together, these models offer a holistic perspective, connecting cognitive distortions and emotional dysregulation to persistent fear [10]. Persistent fear or anxiety can adversely affect patient health, physical condition, and social functioning, making FCR a critical focus for researchers [5].

FCR is influenced by various factors, including gender, anxiety, depression, loneliness, social isolation, and socioeconomic status. However, due to methodological limitations, previous research has generally struggled to comprehensively determine the determinants and consequences of FCR. The Health Ecology Model suggests that individual health outcomes result from the interaction of individual characteristics (e.g., physical and psychological factors), social environments, health policies, and cultural norms [11].

Given that life-threatening hematologic cancers are associated with high mortali-ty rates, patients typically face prolonged and intensive treatments requiring extended hospital stays. As a result of these long-term and intensive treatments, patients fre-quently experience numerous physical challenges and psychological distress [26]. One of these distresses is FCR. FCR has been identified across all cancer types, continents, and time periods following diagnosis. At clinical levels, FCR limits quality of life and daily functioning and requires professional assistance [13]. Identifying hematologic patients at higher risk of experiencing FCR can improve patient management and en-hance the effectiveness of interventions aimed at reducing FCR [27]. However, a lim-ited number of measurement instruments is available to assess FCR among cancer pa-tients in Türkiye yet whether these scales represent valid and reliable measurement in-struments in hematologic cancer patients remains unknown [28]. In this context, ex-amining the psychometric properties of FCR4, FCR6, and FCR7 can help to better un-derstand the concerns of patients experiencing FCR and provide more effective sup-portive care to these patients. Similarly, identifying and supporting individuals with FCR who require intervention can enhance their psychological well-being, improve health outcomes, increase life expectancy, and elevate their overall quality of life [29]. Therefore, this study aimed to adapt FCR4, FCR6, and FCR7 to Turkish and examine their psychometric properties.

Comment#3. Aims can include the population and the center in which the study was conducted. Were adults the only focus here?

Thanks for this helpful comment. We added following to end of Introduction to clarify this.

This study focused specifically on adult patients (aged 18 years and older) with hematological malignancies, as FCR manifestation and measurement may differ be-tween adult and pediatric populations. Therefore, this study aimed to adapt FCR-4, FCR-6, and FCR-7 to Turkish and examine their psychometric properties specifically in adult hematological cancer patients receiving treatment at state hospitals in four different provinces of Türkiye. The multi-center approach was chosen to enhance the generalizability of findings across diverse Turkish healthcare settings and patient demographics. The specific objectives were to examine the construct validity and factor structure of the FCR scales, determine measurement invariance across gender groups, assess convergent and discriminant validity, evaluate known-groups validity across gender, and establish the reliability including both internal consistency and test-retest reliability of the scales in this population.

Comment#4. The description of the study population is more suited for the results portion.

Thanks for this helpful comment. We construct a table in Results section and reported all demographics in this section as below. We also revised Participants section to complement this section.

 Table 1 presents the sociodemographic and clinical characteristics of hematological cancer patients including gender, age, education level, marital status, employment status, disease duration, and disease type. As shown in Table 1, the first sample primarily consisted of males (n = 136, 65.1%) with a mean age of 56.44 years (SD = 14.85), ranging from 18 to 88 years. Participants in this sample were largely elementary school graduates (n = 79, 37.8%), married (n=158, 75.6%), and unemployed (n = 144, 68.9%). Non-Hodgkin lymphoma was the most frequent disease type (n = 88, 42.1%), followed by multiple myeloma (n = 51, 24.4%), Hodgkin lymphoma (n = 41, 19.6%), chronic lymphocytic leukemia (n = 15, 7.2%), and acute myeloid leukemia (n = 14, 6.7%). The mean duration of illness was 50.73 months (SD = 44.44), with a range of 7 to 229 months. The second sample also showed a male predominance (n = 21, 70%) with a mean age of 57.87 years (SD = 14.85), ranging from 20 to 81 years. These participants were mostly high school graduates (n = 10, 33.3%), married (n = 23, 76.7%), and unemployed (n = 23, 76.7%). Multiple myeloma was the predominant diagnosis in the second sample (n = 14, 46.7%), and the mean disease duration was 45.57 months (SD = 44.76), ranging from 8 to 176 months.

Revised Participants section as below.

We conducted this study with two different samples. In the first sample, we per-formed construct, convergent, and discriminant validity analyses as well as item anal-yses of the scales. In the second sample, we conducted test-retest reliability analyses. We selected both samples using convenience sampling methods. A total of 239 patients participated across the two studies. Eligible participants were adults aged 18 years or older with a confirmed diagnosis of hematological cancer. All participants were re-quired to be in remission, receiving treatment at participating hematology clinics, and capable of providing informed consent. Patients were excluded if they had active dis-ease, cognitive impairments that would prevent questionnaire completion, severe psychiatric disorders, or inability to communicate in Turkish.

Comment #5.  The statistical analysis can be split into study outcomes and the analytical methods

Thanks for this helpful comment. We retained the same the section as this is not common in scientific writing. However, we reduced the content as much as possible to  prevent distraction of readers. We changed following sections.

Model 1 tested the adequacy of a single common factor to account for the covariance pattern among FCR-7, FCR-6, and FCR-4 items. Model 1 is consistent with the unidimensional factor structure originally suggested by Humphris et al. [23]. Models 2-5 allowed correlated error terms between specific item pairs (Model 2: items 1-2; Model 3: items 1-7; Model 4: items 4-5; Model 5: items 4-7).

 A non-significant chi-square value indicate excellent fit to data. However, because this value is considerably affected by sample size, researchers note that lower χ2 values indicate better model-data fit. CFI and TLI values ≥0.95 represent excellent fit and 0.90-0.94 indicate acceptable fit. RMSEA values ≤0.05 indicate excellent fit and 0.06-0.10 show acceptable fit. SRMR values ≤0.05 indicate excellent fit and ≤0.08 indicate good fit [42–45].

Following removed:

The parameter a, which represents an item’s discrimination, is classified into the categories of very low, low, moderate, high, and very high.

Comment #6.  Maintain consistency with the citation style

Thanks for helpful comment. We checked our references and corrected all of them as per journal style.  

Comment #7. Has to be cut down in size. Para 2 describes the error terms, but gives a detailed report. Has to be concise to maintain the readers' interest.

Thanks for this helpful comment. We reduced this section as below.

Although researchers consistently support a single-factor structure for FCR scales, they frequently add correlated error terms between specific items [18,19]. In this study, we correlated the error terms of the first and second items because "being afraid" and "being worried/anxious" about cancer recurrence represent highly overlapping emotional responses that participants may interpret almost identically. Thus, their error terms are expected to correlate beyond what the latent FCR factor explains. Future research should test models allowing these error terms to be correlated when examining FCR scale construct validity using confirmatory factor analysis.

Comment #8. A repetition of results is seen in the subsequent paras. Would rephrase these areas.

Thanks for this helpful comment. We read the manuscript two different expert in scale development and reread the article with colleagues. Other reviewer also did not indicate a problem related to repeated content. Although they did not indicated repeat content with results, we tried to minimize similarity with results.

The findings of the convergent and discriminant validity analysis indicate that FCR7, FCR4, and FCR6 scale scores are negatively associated with variables measuring psychological well-being (happiness, general health evaluations, life satisfaction) at low to moderate magnitudes and show moderate positive relationships with variables measuring similar qualities (depression, anxiety, and stress).

The item response theory analysis of the FCR scales also showed high discrimination for most items, with the exception of the sixth and seventh items of the FCR-7 and the sixth item of the FCR-6, which had moderate discrimination. These findings are consistent with previous item response theory analyses of the FCR-7 by Humphris et al. [23].

Similarly, McDonald's omega and Cronbach's alpha reliability values indicate high internal consistency among items across all FCR scales. These findings demon-strate that FCR scales can be used for screening, clinical assessment and decision making, and research purposes. These findings support prior studies that also reported high reliability for the FCR scales [14,15,18–20,23]. Finally, floor and ceiling effects analyses showed participants with extreme scores comprised ≤15% across all FCR scales, indicating absence of problematic effects and providing additional evidence for scale reliability, content validity, and responsiveness [51].

Comment #9. Practical applications are a better fit for the conclusion.

Thanks for this helpful comment. However, this is journal guidelines. As below . https://www.mdpi.com/journal/medicina/instructions

This section is mandatory and should contain the main conclusions regarding the research.

Comment #10. The English could be improved to more clearly express the research.

Thanks for this helpful comment. An American Proofread company  proofread the article.